# Study on Shear Behaviors and Damage Assessment of Circular Concrete Short Columns Reinforced with GFRP Bars and Spiral Stirrups

**DOI:** 10.3390/polym15030567

**Published:** 2023-01-21

**Authors:** Xiaolu Wang, Lingzhu Zhou, Yuke Liang, Yu Zheng, Lixiao Li, Bo Di

**Affiliations:** 1School of Environment and Civil Engineering, Dongguan University of Technology, Dongguan 523808, China; 2Guangdong Provincial Key Laboratory of Intelligent Disaster Prevention and Emergency Technologies for Urban Lifeline Engineering, Dongguan 523808, China; 3College of Civil and Transportation Engineering, Shenzhen University, Shenzhen 518060, China

**Keywords:** shear behaviors, damage assessment, circular concrete short columns, glass fiber-reinforced polymer (GFRP), spiral stirrups

## Abstract

This study investigated the shear resistance and damage evolution of glass fiber-reinforced polymer (GFRP)-reinforced concrete short columns. Five circular concrete short columns reinforced with GFRP bars and spiral stirrups were fabricated and tested under lateral thrust in the laboratory. The test variables involved the stirrup reinforcement ratio, the longitudinal reinforcement ratio and the type of stirrups. The failure modes, load-displacement curves, strain responses and crack characteristics of these columns were documented and discussed. The accuracy of shear design equations in predicting shear capacity of such columns was evaluated. In addition, the digital image correlation (DIC) instrument was used to identify the full-field strain and damage zones of circular concrete short columns. Several smart aggregate (SA) transducers coupled to the surface of these columns were used to monitor its damage status. The energy ratio index (ERI) and the damage index based on smart aggregate were established to characterize damage level of such columns. The test results indicate that the shear capacity is improved 5.6% and 31.1% and the lateral ultimate displacement is increased 67.7% and 400% as the stirrup reinforcement ratio of the concrete short column is increased from 0 to 0.19% and 0.47%, respectively. The shear capacity equation proposed by Ali and his co-workers, considering a strain limit of $0.004E_{fv}$, gives accurate predictions of the shear capacity of circular concrete short columns reinforced with GFRP bars and spiral stirrups. The variation in ERI values is explained by the development of damage zones of the column obtained with DIC technology and with the proposed damage index based on the smart aggregate it is feasible to evaluate the damage level of circular short concrete columns.

## 1. Introduction

Steel materials are prone to corrosion in severe environments, which degrades the mechanical performance of steel-reinforced concrete structures and shortens their service life [1,2,3]. Fiber-reinforced polymer (FRP) material that has the advantages of high tensile strength and corrosion resistance has been advocated for replacing steel materials in bridges and other structures to improve durability [4,5,6]. To solve the problem of performance degradation of a concrete column caused by steel corrosion, some restrained concrete columns with FRP tubes or FRP sheets have been developed [7,8]. In addition, the use of FRP bars and stirrups instead of steel materials in concrete columns has been encouraged [9]; for example, glass-GFRP (GFRP) bars and spiral stirrups have been applied in piers and piles.

Concrete columns are major bearing components in bridges and other structures [10]. Short columns are commonly used in many buildings, such as the piers of viaducts and the basements of tall buildings. Short concrete columns are prone to shear failure under lateral loads induced by earthquakes, wind, ocean waves, traffic, etc. [11], as shown in Figure 1. At present, research on the mechanical behaviors of such concrete columns reinforced with FRP bars and stirrups is limited. Furthermore, damage assessment for short concrete columns is important for providing early warning of possible brittle shear failure. As a result, it is necessary to study the shear behaviors and damage status of concrete short columns reinforced with FRP bars and stirrups.

In recent decades, the shear behavior of concrete beams of rectangular cross-section reinforced with FRP materials has been studied [12,13,14,15,16]. Al-Hamrani et al. [17] has reported on the shear behavior of basalt fiber-reinforced concrete beams reinforced with basalt-FRP (BFRP) bars and glass-FRP (GFRP) stirrups. His study revealed that shear strength and stiffness of concrete beams were significantly reduced if GFRP stirrups were used instead of steel stirrups. The study by Razaqpur et al. [18] verified the accuracy of the shear strength prediction equations in different codes and found that the shear strength of FRP-reinforced concrete members is best predicted by the CSA code. The shear behaviors of circular concrete beams with FRP bars and stirrups have also been investigated [10,11,19]. The research by Mohamed et al. [10,20] indicated that a higher shear capacity was exhibited in circular beams reinforced with carbon-FRP (CFRP) bars and spirals rather than that steel. In addition, Mohamed et al. modified the design equations to better evaluate the shear capacity of circular beams reinforced with FRP spirals. The study by Ali et al. [11,20] showed that the shear capacity of circular beams was improved by using FRP hoops replacing FRP spirals and found that the contribution arch action to shear strength was significant for circular beams with a shear-span ratio less than 2.0. To date, however, the study on the shear resistance of circular concrete columns reinforced with FRP bars and spiral stirrups is rather limited and the shear contribution of FRP spiral stirrups in concrete columns has not been clarified. Moreover, the shear design equations proposed for FRP-reinforced concrete beams with a rectangular cross-section [21,22,23,24] have not yet been proved to be applicable to FRP-reinforced concrete columns with a circular cross-section.

The failure of concrete structures is attributed to the accumulation of damage and a structure’s safety and remaining safe service life can be estimated by the damage level [25]. If the damage initiation of circular concrete columns reinforced with FRP bars and spiral stirrups can be detected, the crack development can often be controlled and brittle shear failure can be avoided. At present, inexpensive yet sensitive piezoelectric transducers with a fast response time are widely used for monitoring the health of structures [26,27,28]. A series of studies has shown that piezoelectric transducers can be used to detect the cracks and evaluate the damage status of a concrete structure [29,30]. Moslehy et al. [31] carried out the damage monitoring of concrete columns under cyclic combined loading using smart aggregate (SA) transducers. His study revealed that a damage index established by the wavelet packet method can accurately reflect the development of damage. Research by Xu et al. has demonstrated the effectiveness of damage monitoring of the concrete columns under blast loads using piezoceramic transducers [32]. Monitoring with piezoelectric transducers can be supplemented by digital image correlation (DIC). It is commonly used to observe full-field strain and identify damage zones [33,34,35]. There has, however, been little published on explaining the variation in piezoelectric signals using the damage zones obtained with DIC. Furthermore, the damage indexes established in numerous studies using piezoelectric transducers are never associated with damage parameters of concrete structures.

In this paper, five circular concrete short columns reinforced with GFRP bars and spiral stirrups were fabricated and subjected to lateral thrust to study their shear behaviors and damage status. The failure modes, load-displacement curves, strain responses and crack characteristics of these columns were studied and discussed. The experimental data were applied to assess the accuracy of existing design equations of shear strength in predicting the shear capacity of circular concrete short columns reinforced with GFRP bars and spiral stirrups. In addition, several smart aggregate transducers were pasted on the surface of these columns to monitor their damage status. DIC was also used to obtain the full-field strain and aggregates were compared with those determined using the load-displacement curves. The development of damage zones identified using DIC explained the changes in the energy ratio index (*ERI*) based on the signal received by the smart aggregate. The damage indexes established using the smart aggregate were compared with those determined using the load-displacement curves.

## 2. Experimental Program

### 2.1. Materials

#### 2.1.1. Concrete Materials

Commercial concrete with coarse aggregate particle size less than 30 mm was used in this test and the design compressive strength was 40 MPa. The standard cubic specimens of size 150 mm× 150 mm × 150 mm were reserved for compressive strength test. The 28-day compressive strength of the test specimens was 41.8 MPa, 44.5 MPa and 49.1 MPa, for an average of 45.1 MPa.

#### 2.1.2. Reinforcing Materials

The GFRP longitudinal bars were provided by Guangdong Dextra Building Materials Co., Ltd. (Guangzhou, China). The GFRP spiral stirrups, which were manufactured by Guangdong Pulwell Composites Co., Ltd. (Zhongshan, China), were adopted in this test. The diameter of the GFRP longitudinal bars was 13 mm and that of the spiral stirrups was 6 mm. In addition, the round steel stirrups with diameters of 6 mm were adopted in this experiment, which are designed as a comparison specimen with the test specimens constructed using GFRP spiral stirrups. The GFRP longitudinal bars (Figure 2a), the spiral stirrups (Figure 2b) and the round steel stirrups (Figure 2c) are illustrated in Figure 2. The tensile properties of GFRP bars were obtained by tensile test according to ACI 440-3R standard [36] and the mechanical properties of the reinforcing materials are summarized in Table 1.

### 2.2. Specimen Design and Fabrication

Five circular concrete short columns were designed, and their parameters are listed in detail in Table 2. The experimental variables were the stirrup reinforcement ratio, the longitudinal reinforcement ratio and the type of stirrup. The GFRP bars were used as longitudinal reinforcement for all concrete short columns. Column G1 was the standard contrast test specimen with a longitudinal reinforcement ratio of 1.5% and a stirrup reinforcement ratio of 0.19%. Column G2 had a higher stirrup reinforcement ratio (0.47%) and column G4 had a higher longitudinal reinforcement ratio (2.25%) than column G1. Column G3 had no stirrups and column S1 used a steel stirrup. All of the concrete short columns were 300 mm in diameter, 400 mm in clear height, 600 mm in lateral force point height and 20 mm in concrete protective layer. Concrete blocks reinforced with steel bars and stirrups were constructed at each end of the concrete short column. The bottom concrete block, which has a size of 1000 mm (length) × 450 mm (width) × 500 mm (depth), served as a fixed end, providing sufficient stiffness to prevent the whole component from overturning during the loading process. The top concrete block, which has a size of 415 mm (length) × 415 mm (width) × 400 mm (depth) was used to apply the load. The ensemble is shown in Figure 3 and the detailed structure of the concrete short column is illustrated in Figure 4. The longitudinal reinforcement ratio was varied by changing the number of longitudinal bars. Their layout in columns G1 and G4 is presented in Figure 5a. The stirrup reinforcement ratio was varied by changing their spacing. That layout in columns G1 and G2 is shown in Figure 5b.

The fabrication of the concrete short columns is shown in Figure 6. Firstly, one end of each longitudinal bar was inserted into a steel sleeve and the expansive cement was poured between the steel sleeve and the longitudinal bar. This aims to ensure that the longitudinal bar was fully anchored in the bottom concrete. Secondly, the longitudinal bars were positioned using steel plate fixtures and the stirrups were fixed along the longitudinal bars at regular intervals. The longitudinal bars were tied together with the stirrups to form a reinforcement cage, see Figure 6a. Subsequently, a PVC tube served as formwork for the column. A cut was reserved in the upper part of the PVC tube, which is designed to ensure the compactness of the column through the outflow of the concrete slurry. The reinforcement cage was placed inside the PVC tube and wooden molds for the top and bottom blocks, see Figure 6b. Finally, the commercial concrete was cast and the formwork was removed when the concrete reached sufficient strength, see Figure 6c. All of the test specimens were watered daily during curing.

### 2.3. Test Setup and Procedure

Ten strain gauges (S1–S10) were attached to each spiral stirrup from the bottom to the top of the column at intervals of 180° (see Figure 7a). These strain gauges are used to assess the level of shear contribution of the stirrups. Three strain gauges (T1–T3, B1–B3) were attached to the longitudinal bars on the tension and the compression sides of the column to track its deformation. In addition, four concrete strain gauges (C1–C4) were uniformly pasted on the surface of concrete along the anticipated direction of the main diagonal crack, which aims to identify the status and deformation of the concrete. As shown in Figure 7b, four linear variable differential transformers (LVTDs) were installed along the column height direction to monitor the lateral displacement. These strain gauges and LVDTs were connected to a TDS 530 acquisition system to collect corresponding data, as shown in Figure 8a.

The lead zirconate titanate (PZT) patch is the major component of a smart aggregate (SA). The PZT patch is first embedded in the copper shell and then embedded in the marble [29]. The copper shell serves as shielding protection and the marble functions as mechanical protection and waterproof agent [29]. As shown in Figure 8b, three SA transducers were arranged at the end, middle and top, respectively, of the column to detect damage. Considering that cracks could occur at the bottom end of the column, which could affect the SA monitoring. Therefore, the SA located at the end of the column was bonded on the bottom concrete block. The instrumentation for the SA transducers is presented in Figure 8b. It includes a data acquisition and a power amplifier. A sine sweep signal with an amplitude of 5 V was selected as the excitation signal. Considering that the environmental noise of most structures is generally within 100 Hz, the frequency of sine sweep signal was selected in the range of 100 Hz–100 kHz. The sampling time and sampling frequency were set to 1 s and 1 MHz, respectively.

As shown in Figure 9, a multi-channel hydraulic servo loading system with a capacity of 500 kN was used to apply the lateral force on the concrete short column in this test. Lateral force was applied to all columns by using load-controlled under constant compression load. Before cracking, the column was lateral loaded with a loading step of 2 kN and a loading rate of 2 kN/min. The loading step was 4 kN after cracking of the concrete columns and 2 kN when approaching the failure of the specimens. The constant compression load was set at 90 kN, which aimed to restrain the upper end. Therefore, the influence of small compressive stress on the shear behavior is ignored in this paper. The data of load, displacement, strain, crack width and piezoelectric signal were recorded after each loading step. In addition, concrete surface of the column was painted with a carrier speckle pattern to serve as a monitoring area for digital image correlation (DIC) (see Figure 9). The three-dimensional DIC equipment took images once per second during the loading process. Those images were helpful for analyzing the changes in displacement, strain and cracks of these columns.

## 3. Results and Discussion of Shear Behaviors

### 3.1. Crack Patterns and Failure Mode

Figure 10 shows the crack patterns observed in the tests. The test results of all concrete short columns are presented in Table 3. The first crack appeared at the bottom end of these columns, except for column S1 reinforced by round steel stirrups. The load corresponding to the first crack was between 24 and 36 kN. The second and third cracks appeared in the upper or lower part of these columns, and the second crack corresponding load was between 42 and 59 kN. The third crack appeared at loads between 60 and 70 kN. However, notably, the third crack in column S1 appeared only on the other side. Before the load reached 75 kN, the first, second and third cracks developed substantially perpendicular to the height of the column. On increasing the applied load, these horizontal cracks propagated toward the concrete compression zone at the foot of the column and formed a main diagonal shear crack and several branch cracks. The shear crack then propagated to both ends and the concrete in the column’s compression zone was crushed, leading to failure. Such shear-compression failure was observed in all of the tested columns.

It can be seen in Table 3 that the cracking loads of the standard contrast column (G1) and column reinforced by round steel stirrups (S1) are lower than that of the column without stirrups (G3). However, the increase of stirrup reinforcement and longitudinal reinforcement ratio (in G2 and G4) can improve the cracking load compared with the test specimen G1. The shear capacity of test specimen G3 without stirrups is 90 kN. G1 with a stirrup reinforcement of 0.19% and G2 with a stirrup reinforcement of 0.47% have a shear capacity of 95 kN and 118 kN, respectively. Compared to column G3, the shear capacity of column G1 and G2 is increased by 5.6% and 31.1%, respectively. Increasing the stirrup reinforcement ratio restricts the propagation of diagonal cracks, which is equivalent to improving the shear contribution of concrete. The shear capacity of the column is improved by 18.9% as the longitudinal reinforcement ratio increases from 1.5% to 2.25%. The increase in longitudinal reinforcement leads to an enhanced dowel effect, which improves the shear capacity of the column. Compared to a column reinforced with GFRP spiral stirrups (G1), a slightly higher shear capacity of 13.7% occurs in the column reinforced with round steel stirrups (S1). As the elastic modulus of the steel stirrups is larger than that of the GFRP spiral stirrups, which results in stronger radial restraint of steel stirrups on the column, and thus improves the shear capacity of the column. The ratio of cracking load to shear capacity of all concrete short columns is distributed between 22.2% and 36.7%. The effects of the stirrup reinforcement ratio, the longitudinal reinforcement ratio and the stirrup type on shear capacity of tested columns are shown in Figure 11.

### 3.2. Load-Displacement Curves

The load-displacement curves on the top of the concrete short columns obtained with LVDT (D2), as shown in Figure 7b, and DIC are shown in Figure 12. It can be seen that the load-displacement curves obtained with LVDT are almost consistent with that obtained with the DIC instrument. All the load-displacement curves can be divided into three phases. The first phase is from the start of loading until the cracking load. The displacement develops slowly in this stage due to the higher lateral rigidity of the concrete column. The second phase is from the cracking load to about 75 kN which is the development and propagation of flexural cracks. The stiffness of the column is decreased with the appearance of the initial crack and then the displacement is increased almost linearly by increasing the applied load in this phase. The third phase is during the formation and development of the main diagonal shear crack and it ends with the failure of the column. Each small load increment significantly results in larger displacement in this stage.

It can be seen in Figure 12 that the load-displacement curves in the first and second phase (before 75 kN) are barely affected by the stirrup reinforcement ratio, the longitudinal reinforcement ratio and the type of stirrups. While the stiffness of load-displacement curves in the third stage is enhanced by increasing the longitudinal reinforcement ratio, see Figure 12b. On increasing the stirrup reinforcement ratio from 0% to 0.19% and 0.47%, the displacement corresponding to the shear capacity, namely the ultimate displacement, of these columns is increased by 67.7% and 400.0%, respectively; see Table 3 and Figure 12a. The displacement corresponding to the shear capacity is improved by 50.0% when the longitudinal reinforcement ratio of a column increases from 1.5% to 2.25%, as shown in Figure 12b. This indicates that the ductility of the column is improved by increasing the stirrup and the longitudinal reinforcement ratios, especially the stirrup reinforcement ratios. As expected (see Figure 12c), the displacement corresponding to the shear capacity of the column reinforced by GFRP spiral stirrups (G1) is slightly smaller than that of the column reinforced by round steel stirrups (S1). On the whole, however, the load-displacement curves of the column reinforced by GFRP spiral stirrups (G1) and round steel stirrups (S1) are similar. This indicates that it is feasible to use GFRP spiral stirrups instead of round steel stirrups in concrete short columns. The effects of the stirrup reinforcement ratio, the longitudinal reinforcement ratio and the stirrup type on ultimate displacement of tested columns are shown in Figure 13.

The lateral displacements of the columns from the bottom to the top obtained by DIC are shown in Figure 14. Prior to the formation of the main diagonal cracks (that is, before about 75 kN for these columns), the displacement is almost linearly distributed along the direction of the column height. Afterwards, the lateral displacement is increased significantly due to the formation and propagation of the main diagonal crack. The lateral displacement along the column height is then no longer linearly distributed, because the column is broken into multiple concrete blocks by the diagonal cracks.

### 3.3. Strain Responses

#### 3.3.1. Concrete Strain

It is difficult to detect the strain at the main diagonal crack by using strain gauges attached to the concrete surface. Therefore, the strain corresponding to the acquisition point at the main diagonal crack (see Figure 15) is extracted by using the DIC technique. The effects of different parameters on the concrete strain are shown in Figure 15. It can be seen that the concrete strain values are at a small level prior to the onset of the main diagonal crack. Then, the concrete strain values are increased rapidly with the formation of the main diagonal crack. The load corresponding to the abrupt change in concrete strain values is improved by increasing stirrup and longitudinal reinforcement ratios (see Figure 15a,b). This indicates that the increase in stirrup and longitudinal reinforcement ratios can effectively inhibit the formation of a main diagonal crack. Figure 15c shows that the load corresponding to the abrupt change in concrete strain values for specimen G1 is almost the same as that for specimen S1 reinforced with round steel stirrups. This shows that the load corresponding to the formation of a main diagonal crack is barely affected by the type of stirrups.

#### 3.3.2. Reinforcement Strain

Figure 16 presents the tensile and compressive strains of the longitudinal reinforcement. The maximum tensile and compressive strains occur at the bottom of the longitudinal reinforcement. Both the tensile and compressive strains in the longitudinal reinforcement are insignificant before the concrete cracks. As the applied load increases, the tensile strains at T3, T2 and T1 develop inflection points in turn, after which they increase at a faster rate. The inflection points mean that the tensile stress provided by the concrete is transferred to the longitudinal tensile reinforcement due to the occurrence of flexural cracks. Strain gauge T3 at the bottom of the longitudinal reinforcement is easily damaged, caused by the generation of cracks. Therefore, the strain gauges T2 and B3 are selected to analyze the contribution level of longitudinal reinforcement of the columns with different parameters (see Figure 17). It can be observed from Figure 17a,b that the tensile and compressive strains increase by increasing the applied load, and this increase is mitigated after the occurrence of flexural cracks when a longitudinal reinforcement ratio of 2.25% and a stirrup reinforcement ratio of 0.47% are used. Figure 17b indicates that the contribution level of the longitudinal reinforcement after the occurrence of flexural cracks is weakened by increasing the stirrup reinforcement ratio. As shown in Figure 17c, the tensile strain in the longitudinal reinforcement at loads greater than 60 kN is greater in column G1 than column S1. Compared with GFRP spiral stirrups, the steel stirrups with higher elasticity modulus can better restrain the development of diagonal cracks and reduce the lateral deformation.

Although multiple strain gauges are used to record strains in the stirrups, the main diagonal crack does not always cross the stirrups where the strain gauges are located. The larger strain values recorded by two gauges near the main diagonal crack are therefore selected for analysis, as shown in Figure 18. Stirrup strain is small before the main diagonal crack formed, which means that the shear resistance is mainly provided by the concrete. Thereafter the stirrup strain is increased sharply, indicating that the internal force is transferred from the concrete to the stirrups and the stirrups are providing more of the shear resistance. Figure 18a shows that stirrup strain is reduced by increasing the stirrup reinforcement ratio after the formation of the main diagonal crack. This can be attributed to the fact that the shear force is distributed over a larger number of stirrups. Increasing the longitudinal reinforcement ratio also decreases the strain in the stirrups after the formation of the main diagonal crack (see Figure 18b). Since the increase in the number of longitudinal bars leads to an improvement in the shear force carried by the dowel action and a decrease in the shear force carried by the stirrups under the same load. The values of stirrup strain in column G1 are greater than that of column S1 at the stage of approaching failure, as shown in Figure 18c. This could be attributed to the fact that the steel stirrups have higher elastic modulus that restricts the development of the main diagonal crack.

### 3.4. Crack Development

Figure 19, Figure 20 and Figure 21 display principal strain nephograms of test specimens (G1, G3, S1) under various loading steps obtained by using DIC. The principal strains in the areas monitored by DIC ranged from 0 to 10,000 micro-strains. In the figures, the regions where the strain is close to 0 micro-strain are shown in purple and the regions where the strain approaches 10,000 micro-strains are presented in red. The red regions roughly correspond to the damage zones of the concrete surface. It can be seen from Figure 19, Figure 20 and Figure 21 that, as would be expected, the number and length of damage zones are increased by increasing the applied load. The distribution and development of damage zones are consistent with the observed cracks (see Figure 10), which indicates that the propagation of cracks can be reflected by the damage zones. The first cracks in columns G1 and G3 are at the bottom of the column where they could not be detected with DIC in this setup. Since the damage zone at the bottom of the column is not in the areas monitored with DIC.

As shown in Figure 19, Figure 20 and Figure 21, the initial flexural crack consistently occurred in the bottom of a column. Increasing the applied load results in the propagation of an initial flexural crack and the formation of additional flexural cracks. The flexural cracks deflect at an applied load of about 80 kN and begin to form flexural-shear diagonal cracks. Further increase in the applied load promotes the development of those flexural-shear diagonal cracks, forming the main diagonal shear crack. The main diagonal shear crack eventually propagates through the entire cross-section of the column, causing its failure.

### 3.5. Theoretical Analysis of Shear Capacity

#### 3.5.1. Shear Strength Contribution of FRP Spiral Stirrups

The shear strength of a column without stirrups is mainly provided by the shear resistance of uncracked concrete, dowel action between the longitudinal bars and the concrete, residual tensile stress and interlock effect between aggregates at the diagonal cracks. Moreover, the contribution of an arch effect to shear strength cannot be ignored for these members with a shear-span ratio less than 2.5 [37]. In a column with spiral stirrups, apart from the above concrete shear contribution components, spiral stirrups provide additional shear resistance. Furthermore, using spiral stirrups in the concrete column can improve several shear contribution components of concrete. This is attributed to the following: (1) The use of spiral stirrups can inhibit the development of diagonal cracks, thus increasing the contribution of aggregate interlock to shear strength. (2) The position locking of the spiral stirrups to longitudinal bars can improve the contribution of dowel action to shear strength. (3) The binding of spiral stirrups with the longitudinal bars forms a stable cage, which can effectively restrain the concrete in the core area and improve its strength, thus improving the contribution of uncracked concrete in core area to shear strength.

The efficiency of spiral stirrups in resisting shear force is reduced compared with the discrete hoop stirrups. Only a component of the force in the helical link can resist shear force resulting from the spiral inclination and geometric curvature of spiral stirrups (see Figure 22). The shear strength provided by the spiral stirrups ($V_{f}$) can be estimated by introducing two factors, $\lambda_{1}$ and $\lambda_{2}$, representing the geometric curvature of the spiral stirrup and the spiral inclination [38]. Combined with the traditional truss mode, the effective stress of the spiral stirrup ($f_{fv}$) of the test specimen at failure can be expressed as follows:(1)$$f_{fv}=\frac{\left(V_{exp}-V_{c}\right)s}{\lambda_{1}\lambda_{2}A_{fv}d\cot\theta }$$
where $V_{exp}$ refers to the shear strength of the test specimen at failure; $V_{c}\;$ is the shear strength provided by the concrete; θ means the angle of inclination of main the diagonal crack; $A_{fv}$ represents the cross-sectional area of the spiral stirrup; *s* and *d* refer to the pitch of spiral stirrups and the effective depth, respectively. In this calculation, in accordance with reference [10], the values of $\lambda_{1}$ and $\lambda_{2}$ are taken as 0.85 and 1.0, respectively. The *d* for circular concrete members can be determined according to *AASHTO LRFD bridge design specifications* [39], as below:(2)$$d=\frac{D}{2}+\frac{D_{r}}{\pi }$$
where *D* is the diameter of the circular cross-section and *D_r_* is the diameter of the circle passing through the centers of the longitudinal reinforcement. It is worth noting that the applied vertical compression load is relatively small in this study, so its influence on shear strength is ignored.

In this experiment, the shear resistance is considered to be provided by the concrete when the strain of the spiral stirrup is less than 100 micro-strains. That enables determining $V_{c}$. The angle of inclination of the main diagonal crack is not easily measured, so 45° was assumed in this study. Figure 23 shows the ratio of the effective stress of spiral stirrups ($f_{fv}$) calculated according to Equation (1) to the ultimate tensile strength ($f_{fu}$) of a GFRP straight bar. It can be observed that the effective stress of the spiral stirrups is 23.5%, 20.1% and 26.0% of the ultimate tensile strength of a GFRP straight bar for columns G1, G2 and G4, respectively. The effective stress of the spiral stirrup is decreased with the increase in the stirrup reinforcement ratio, while it is improved by increasing the ratio of longitudinal tensile reinforcement. The effective stress of the spiral stirrups is compared with the limit of two specifications (see Figure 23). It can be found that the average effective stress of the spiral stirrups in the test specimens is $0.0043E_{fv}$, $E_{fv}$ is the elasticity modulus of GFRP stirrup. That exceeds the ACI strain limit ($0.004E_{fv}$) [21], but is lower than the CSA stress and strain limits ($0.4f_{fu}$ and $0.005E_{fv}$) [22]. Accordingly, the shear strength provided by the spiral stirrups in a column can be predicted relatively accurately using a strain limit of $0.004E_{fv}$.

#### 3.5.2. Prediction and Comparison of Shear Capacity

Currently, existing specifications have not been used to evaluate the shear capacity of circular concrete columns reinforced with FRP bars and spiral stirrups. In this section, four shear design specifications which might be applied to predict the shear capacity of the concrete columns reinforced with GFRP bars and spiral stirrups are summarized in Table 4. The detailed equations of shear capacity of FRP-reinforced concrete members are composed of $V_{c}$ and $V_{f}$. $V_{c}$ depends on the axial compressive strength of concrete, the longitudinal tensile reinforcement ratio, the effective depth, the width of the section and other factors. However, only the CSA code takes into account the shear–span ratio in predicting $V_{c}$. The expressions for calculating $V_{f}$ are similar in these specifications, with the main difference being the limit value of $f_{fv}$. In addition, the angle of inclination of the main diagonal crack is considered in the CSA code and the angle of inclination of the spiral stirrups is employed in the ACI and JSCE formulations for calculating $V_{f}$.

Ali et al. [11] recommended introducing $\lambda_{1}$, $\lambda_{2}$ and $\lambda_{s}$ to calculate the shear strength provided by FRP spiral stirrups, as follows:(3)$$V_{f}=\lambda_{1}\lambda_{2}\lambda_{s}\frac{A_{fv}f_{fv}d_{v}}{s}\cot\theta$$
where $\lambda_{s}$ is a strength reduction, taken as 0.85 for GFRP spiral stirrups. $d_{v}$ is the effective height of concrete members, $d_{v}=0.9d$. Based on the findings in Figure 23, it is recommended that the effective stress of the GFRP spiral stirrup ($f_{fv}$) be considered to be $0.004E_{fv}$. In addition, the shear strength offered by the concrete can be accurately predicted by the design equation in the CSA code considering the arch effect. Therefore, the shear capacity of a circular concrete short column reinforced with FRP bars and spiral stirrups can be calculated by combining Equation (3) with $V_{c}$ in the CSA code. It provides more reasonable and accurate prediction results than other specifications (see Table 5 and Figure 24), with an average value of experimental to predicted shear capacity ratio of 0.94. Therefore, the shear capacity equation proposed by Ali et al. is more precise with regard to predicting the shear capacity of circular concrete short column-reinforced GFRP bars and spiral stirrups.

## 4. Damage Assessment and Discussion Based on Piezoceramic Transducers

### 4.1. Damage Detection Principles of Concrete Short Column Enabled Active Sensing

In this study, several smart aggregate (SA) transducers were used to monitor the crack development and evaluate the structural damage of the concrete short columns. The active sensing technology that is, using a pair of SAs, one as the actuator and the other as the sensor, was adopted [40,41]. The stress waves were generated by the SA acting as actuator and are received by the SA acting as sensor. As shown in Figure 25, SA1 acts as an actuator and SA3 acts as a sensor, which aims to monitor the overall structural damage of the column. In addition, using SA1 as the actuator and SA2 as the sensor can detect the damage of the lower part of the column. Similarly, the damage of the upper part of the column can be detected by using SA2 which functions as an actuator and SA3 which functions as a sensor.

In Figure 25a, with a healthy state in the column, the stress wave propagates along the concrete structure barely with attenuation. As shown in Figure 25b, with a crack in the lower part of the column, the propagation of stress wave generated by SA1 is impeded by the crack, resulting in the attenuation of the stress wave received by SA2 and SA3. However, the stress wave generated by SA2 and received by SA3 has little attenuation, since no cracks are present in the upper part of the column. When a crack appears at the upper part of the column (see Figure 25c), the stress wave generated by SA1 is received by SA2 with small attenuation. It is received by SA3 with significant attenuation due to the obstruction of the crack. Therefore, the cracks can be localized and the damage status of the overall structure can be evaluated according to the variation in the received stress wave.

### 4.2. Time-Domain and Frequency-Domain Analysis

Time-domain signals of the overall structure, that is, the stress wave generated by SA1 and received by SA3, are presented in Figure 26. After the column is cracked, the increased load results in a decrease in the amplitude of the time-domain signal. This is caused by the generation and development of cracks that hinder the propagation of stress waves. It also indicates that the damage level of the columns is constantly increasing. When the columns are close to the failure status, the received time-domain signals are roughly on the level of the noise in the signals. This indicates that the propagation of the stress wave is mostly impeded due to cracks penetrating the entire column cross-section (see Figure 21).

Frequency-domain diagrams corresponding to the time-domain signals in Figure 26 are shown in Figure 27. It can be found that the amplitude of frequency-domain diagrams shows a decreasing trend by increasing the load when the column is in a cracked state. For the column without stirrups (see Figure 27b), two peaks with similar amplitude are presented, corresponding to the frequencies of about 19 kHz and 68 kHz. The configuration of the GFRP spiral stirrups in the column, as shown in Figure 27a, causes the first peak to shift to the right and the second peak to move to the left. The amplitude of the second peak is smaller than that of the first peak and the two peaks correspond to frequencies of about 21 kHz and 63 kHz, respectively. For the column with round steel stirrups (see Figure 27c), there is only one peak, which corresponds to a frequency of 32 kHz. That appears to be obtained by merging the first and second peaks of the column without stirrups. This means the first peak is shifted to the right and the second peak is moved to the left due to the configuration of round steel stirrups. The above analysis shows that the frequencies corresponding to the peaks are varied with the configuration of the stirrups in the column.

### 4.3. Energy Ratio Index (ERI) Analysis Based on Wavelet Decomposition Method

Wavelet decomposition is a common method of signal processing in structural health monitoring [42]. In this study, wavelet decomposition is used to decompose the original signal (*S*) into two parts: a detailed signal with a high frequency (*D*_1_) and an approximation signal with a lower frequency (*A*_1_). The approximation signal (*A*_1_) is then further decomposed into a second-level detailed signal with a higher frequency (*D*_2_) and a second-level approximation signal with a lower frequency (*A*_2_), and so on. Finally, the original is decomposed into *D*_1_, *D*_2_, *D*_3_, …, *D_n_*_−1_, *D_n_* and *A_n_* after *n* rounds of decomposition, termed *n*-level wavelet decomposition (see Figure 28). It can be observed from the process of *n*-level wavelet decomposition that only the obtained approximation signal is decomposed, which indicates that wavelet decomposition is more suitable for processing signals with low-frequency information. In addition, detailed information in different frequency bands can be extracted through wavelet decomposition.

In *n*-level wavelet decomposition, the original signal (*S*) is divided into *n*+1 signals corresponding to different frequency bands. Those signals in different frequency bands can be represented by a set *X_j_*, which can be expressed as:(4)$$X_{j}=\left[X_{j,1},X_{j,2},X_{j,3}\ldots X_{j,m-1},X_{j,m}\right]$$
where *j* represents the frequency bands (*j* = 1, 2, 3, … *n*−1, *n*, *n*+1) and *m* represents the amount of sampling data. The energy of those signals in different frequency bands can be obtained as:(5)$$E_{j}={\left\|X_{j}\right\|}_{2}^{2}=\sum_{k=1}^{k=m}X_{jk}^{2}=X_{j,1}^{2}+X_{j,2}^{2}+X_{j,3}^{2}+\ldots X_{j,m-1}^{2}+X_{j,m}^{2}$$

In this experiment, the signal collected each time is denoted as *S_i_* and the corresponding energy in different frequency bands obtained by the *n*-level wavelet decomposition can be expressed as *E_i,j_*. Therefore, an energy ratio index in the different frequency bands (*ERI _j_*) can be defined as:(6)$$ERI_{j}=\frac{E_{i,j}}{E_{1,j}}$$
where *E_i,j_* represents the energy of the signal collected at the *i*th time in the *j*th frequency band. *E*_1,*j*_ is the energy of the first acquisition signal in the *j*th frequency band, which corresponds to the time of the member being in a healthy state. Further, an energy ratio index (*ERI*) can be defined as:(7)$$ERI=\frac{\sum_{j=1}^{j=n+1}E_{i,j}}{\sum_{j=1}^{j=n+1}E_{1,j}}$$

In this study, 3-level wavelet decomposition is applied to all the signals collected. The high-frequency component was the noise, which is removed in calculating the *ERI*.

The variation in *ERI* with load is presented in Figure 29. Among them, SA1-SA3 refers to the stress wave generated by SA1 and received by SA3. It can be seen that the *ERI* values barely vary before the onset of cracking of the columns, but they then drop significantly afterward. The change in *ERI* values relates to the damage zones, that is, the generation and propagation of cracks, as shown in Figure 19, Figure 20 and Figure 21. Take the test specimen G1 as an example: (i) The first crack is observed at the bottom of the column with the load at about 24 kN, as shown in Figure 19a. Correspondingly, the *ERI* values for SA1-SA3 and SA1-SA2 begin to drop at about 21 kN, see Figure 29a, with the appearance and development of the first crack. Meanwhile, the first crack at the bottom of the column has scarcely an effect on the *ERI* values for SA2-SA3. (ii) As the load increases to 51 kN, the second crack is observed in the upper part of the column; see Figure 19c. Correspondingly, the *ERI* values for SA2-SA3 decrease significantly from 40 kN for SA2-SA3 in Figure 29a. (iii) Further increase in the load leads to the rapid development of the first and second cracks and even the appearance of a third crack. The *ERI* values decline to very small, since the propagation of stress waves is mostly impeded by the long and wide cracks. Note that for the test specimen S1, the *ERI* values for SA1-SA2 and SA2-SA3 are almost zero after the second crack forms. This is owing to the fact that the second crack formed just at the position of SA2. The above analysis illustrates that the loads corresponding to the first crack in the upper and lower parts of a column can be identified using the *ERI* values for SA1-SA2 and SA2-SA3, respectively. The identification results of crack formation are consistent with those observed by the naked eye (see Table 3) and monitored with the DIC (see Figure 19, Figure 20 and Figure 21), though both generally lag in *ERI* identification. This is because the micro-cracks may not be visible to the naked eye and the sides of a column are blind spots for DIC measurement (see Figure 9).

### 4.4. Damage Assessment

In recent years, piezoceramic transducers have been widely used to monitor concrete structures. This has led many scholars to propose damage indices to characterize the damage level of concrete members [43,44]. Among them, the root-mean-square deviation (*RMSD*) based on wavelet energy is a classic damage index [45]. Its calculation is rather complicated, so this study defines a damage index based on the *ERI* values to evaluate the damage level of concrete structures, as follows:(8)$$DI_{SA}=1-ERI$$

In this section, only the signals for SA1-SA3 are used to obtain the damage indices to quantify overall damage of the column.

In general, the damage level of a reinforced concrete column can be reflected by the stiffness variation of the load-displacement curve. A damage index based on the load-displacement curve can be defined as shown below:(9)$$DI_{P-\delta }=1-\frac{P/\delta }{P_{cr}/\delta_{cr}}$$
where *P* and *P_cr_* are the applied load and the cracking load, respectively; *δ* and *δ_cr_* are the displacements corresponding to the applied and cracking loads. In this study, the damage indices can be calculated using the load-displacement curves at the top of a column monitored with the LVDT and DIC instruments.

The damage indices determined according to Equations (8) and (9) are compared in Figure 30. The damage indices determined from the SA agree well with those obtained from load-displacement curves monitored with the LVDT and DIC. This indicates that damage indices obtained from SA can be used to evaluate the damage level of the columns. Before the column cracks, the damage indices are almost 0, indicating that the member is in a healthy state. After the first crack forms, the damage indices increase roughly linearly for test specimen G3 (column reinforced without stirrups), but the damage indices increase sharply and then slowly for test specimens G1 (column reinforced with GFRP spiral stirrups) and S3 (column reinforced with round steel stirrups).

## 5. Conclusions

In this study, the thrust test was carried out on five circular concrete short columns reinforced with glass fiber-reinforced polymer (GFRP) bars and spiral stirrups under constant compression load to investigate their shear behaviors. The effects of three test parameters (the stirrup reinforcement ratio, the longitudinal reinforcement ratio and the type of stirrups) on load-displacement curves, strains, cracks and failure modes were discussed. Several design equations for predicting the shear capacity of circular concrete short columns reinforced with GFRP bars and spiral stirrups were compared and evaluated. In addition, several smart aggregate (SA) transducers were used to assess the damage levels of circular concrete short columns. Based on the above studies, the following conclusions can be drawn:GFRP spiral stirrups and longitudinal reinforcement can inhibit the formation of the main diagonal crack, increase the shear capacity and improve the ductility of a concrete short column. The load-displacement curve of a short column reinforced by GFRP spiral stirrups is similar to that of one reinforced by round steel stirrups, indicating that it is feasible to use GFRP bars and GFRP spiral stirrups in column constructions.GFRP spiral stirrups increase the shear capacity of a circular concrete short column by 5.6% and 31.1% at a stirrup reinforcement ratio of 0.19% and 0.47%, respectively. The shear capacity of a concrete short column with GFRP spiral stirrups is improved by 18.95% by increasing the longitudinal reinforcement ratio from 1.5% to 2.25%. Notably, the lateral ultimate displacement is increased by 67.7% and 400% at a stirrup reinforcement ratio of 0.19% and 0.47% in a circular concrete short column.The effective stress of the GFRP spiral stirrups at failure, which is from 20.1% to 26.0% of the ultimate tensile strength of a GFRP straight bar, is decreased with the increase in the stirrup reinforcement ratio and is improved by increasing the longitudinal reinforcement ratio.For circular concrete short columns with GFRP bars and spiral stirrups, the shear capacity can be accurately predicted by using the equation proposed by Ali et al., with a strain limit of $0.004E_{fv}$. The CSA code provides relatively accurate shear capacity predictions and the ACI, JSCE and BISE codes provide relatively conservative shear capacity predictions.The energy ratio index (ERI) established by wavelet decomposition can help to identify the location of cracks in a column and the loads corresponding to the formation of cracks. Damage indices determined using smart aggregate are consistent with those obtained from load-displacement curves, indicating that with the proposed damage index based on the smart aggregate it is feasible to evaluate the damage level of the concrete short columns.

## Figures and Tables

**Figure 1 polymers-15-00567-f001:**
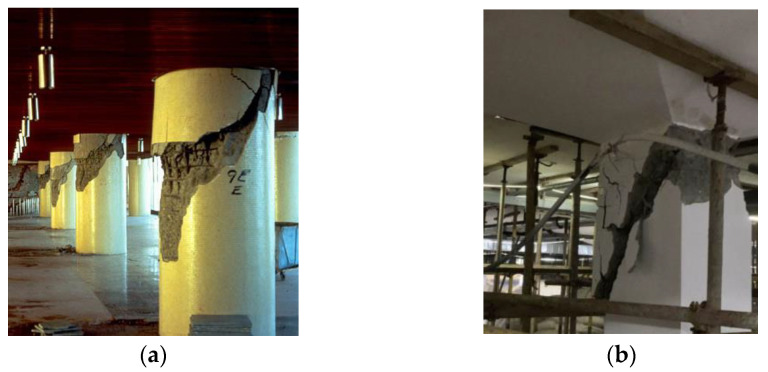
Shear failure of concrete short columns. (**a**) Circular concrete short columns, (**b**) Cubic concrete short columns.

**Figure 2 polymers-15-00567-f002:**
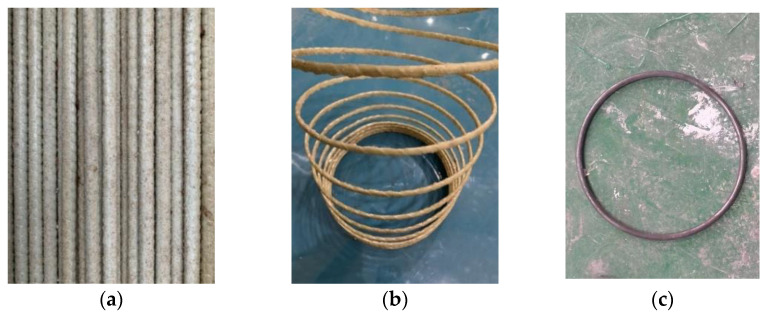
Longitudinal bars and stirrups. (**a**) GFRP longitudinal bars, (**b**) GFRP spiral stirrups, (**c**) Round steel stirrups.

**Figure 3 polymers-15-00567-f003:**
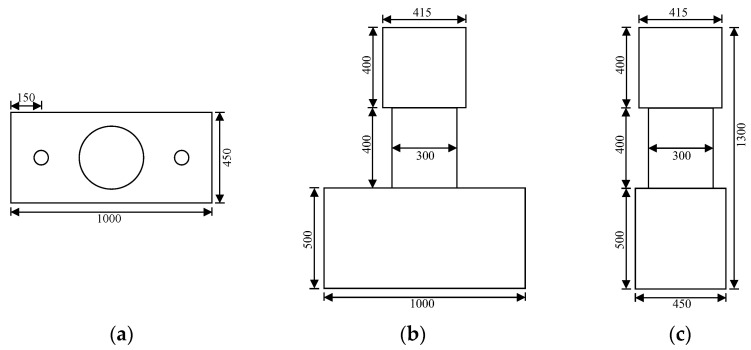
The size of a concrete short column and its concrete blocks. (**a**) Top view, (**b**) Front view, (**c**) Side view.

**Figure 4 polymers-15-00567-f004:**
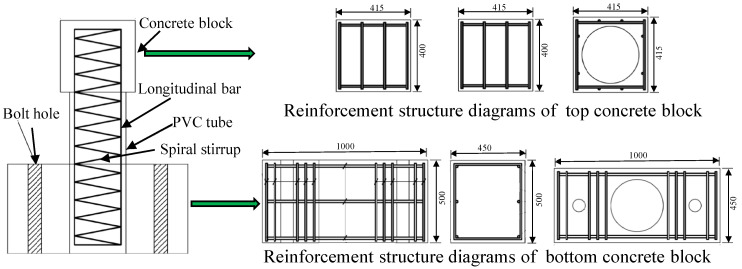
The structure diagrams of a concrete short column in detail.

**Figure 5 polymers-15-00567-f005:**
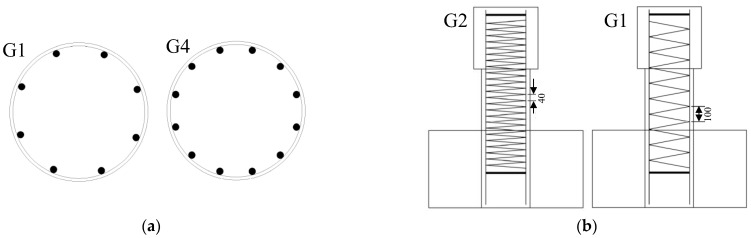
The structure of the concrete short columns in detail. (**a**) Longitudinal bars, (**b**) Spiral stirrups.

**Figure 6 polymers-15-00567-f006:**
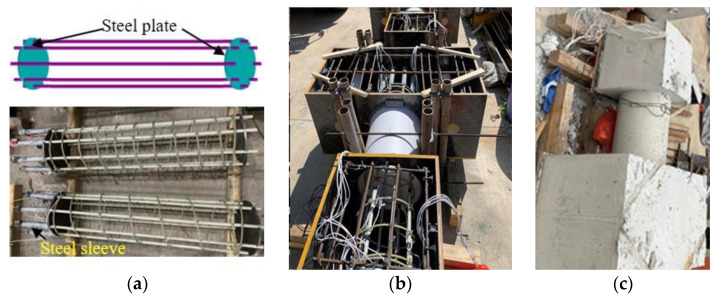
Fabrication of the concrete short columns. (**a**) Positioning of longitudinal bars, (**b**) Before the casting, (**c**) After the casting.

**Figure 7 polymers-15-00567-f007:**
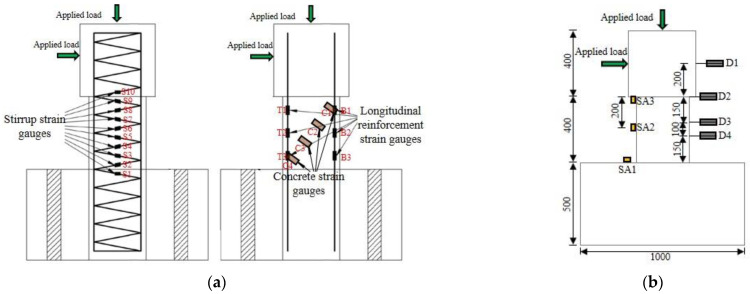
Arrangement of the transducers. (**a**) Strain gauges, (**b**) Smart aggregate and displacement transducers.

**Figure 8 polymers-15-00567-f008:**
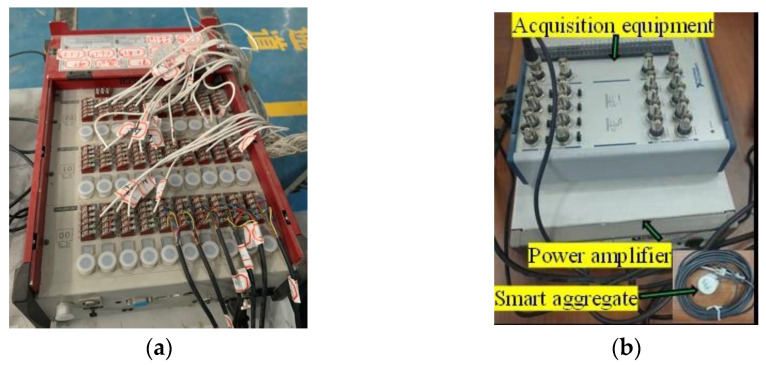
Test instrumentation. (**a**) The instrumentation for the strain gauges and LVDTs, (**b**) The instrumentation for SA transducers.

**Figure 9 polymers-15-00567-f009:**
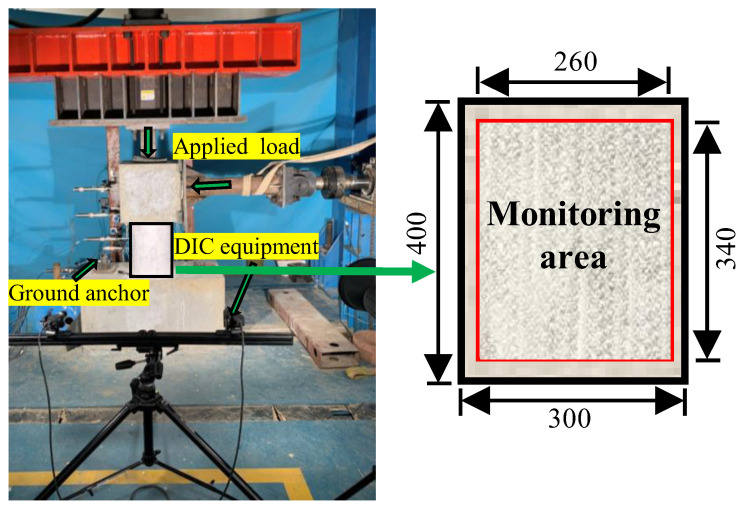
Loading configuration of test specimen.

**Figure 10 polymers-15-00567-f010:**
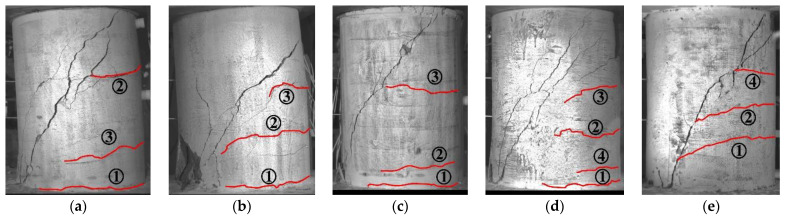
Crack patterns of the concrete short columns. (**a**) Specimen G1, (**b**) Specimen G2, (**c**) Specimen G3, (**d**) Specimen G4, (**e**) Specimen S1.

**Figure 11 polymers-15-00567-f011:**
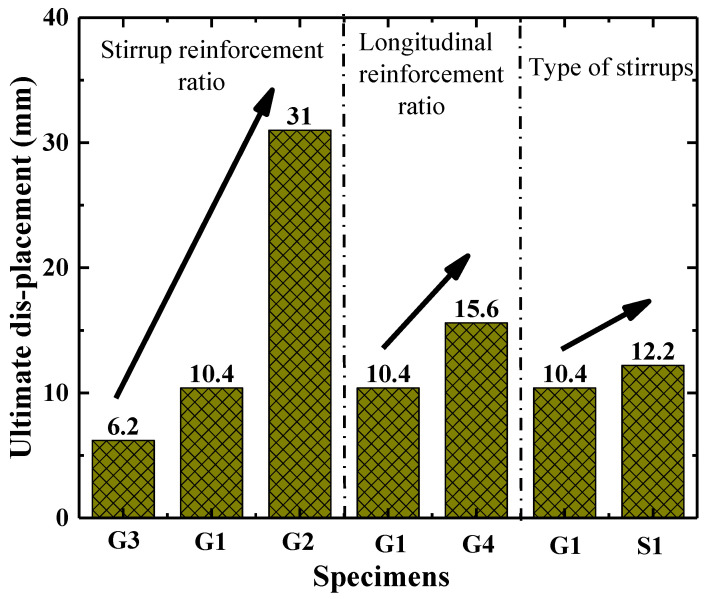
The effect of various parameters on shear capacity.

**Figure 12 polymers-15-00567-f012:**
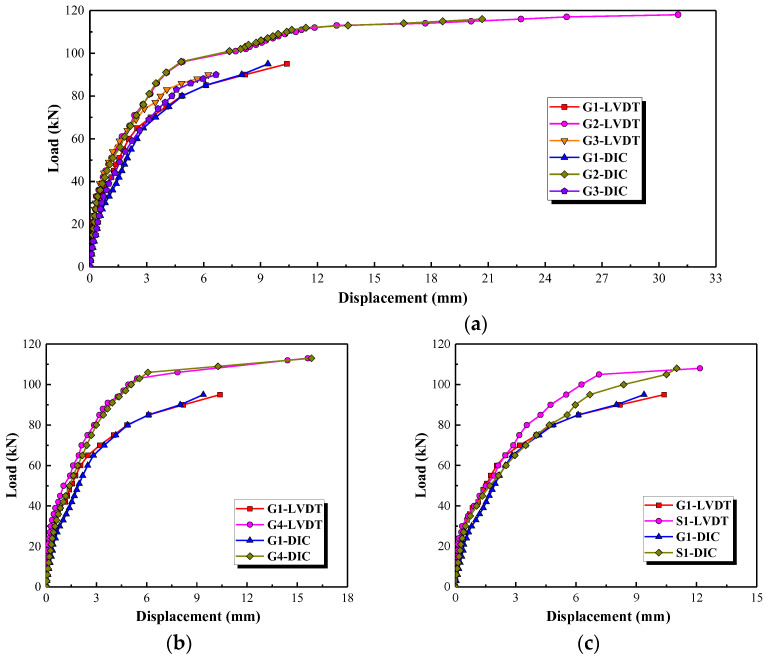
The effect of three parameters on the load-displacement curves. (**a**) Stirrup reinforcement ratio, (**b**) Longitudinal reinforcement ratio, (**c**) Type of stirrups.

**Figure 13 polymers-15-00567-f013:**
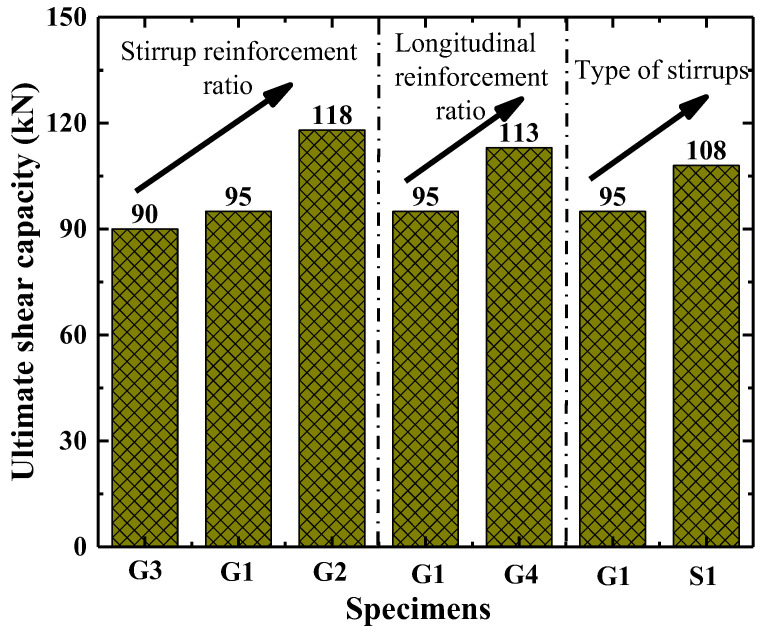
Shear capacity comparisons.

**Figure 14 polymers-15-00567-f014:**
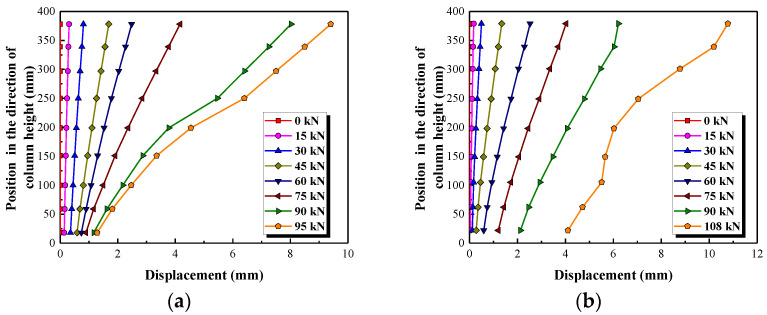
The lateral displacement of the columns obtained by DIC. (**a**) Test specimen G1, (**b**) Test specimen S1.

**Figure 15 polymers-15-00567-f015:**
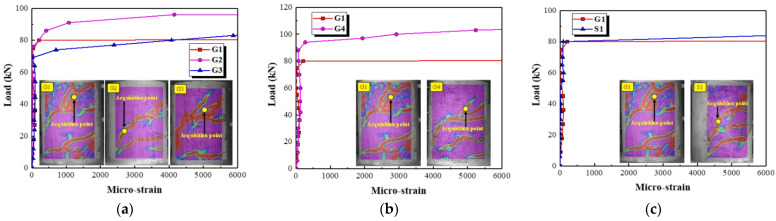
The effect of different parameters on the concrete strain. (**a**) Stirrup reinforcement ratio, (**b**) Longitudinal reinforcement ratio, (**c**) Type of stirrups.

**Figure 16 polymers-15-00567-f016:**
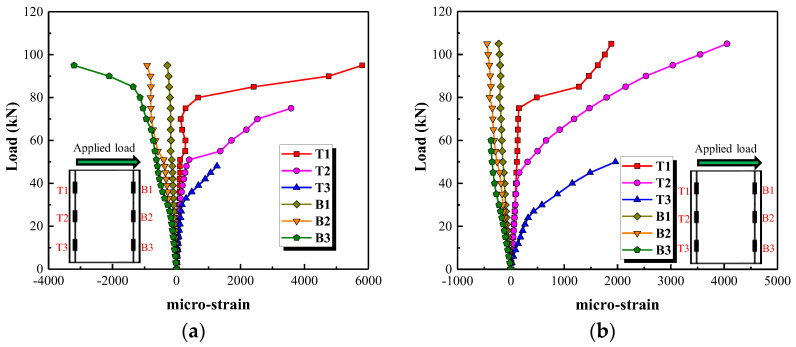
Strain in the longitudinal reinforcement. (**a**) Test specimen G1, (**b**) Test specimen S1.

**Figure 17 polymers-15-00567-f017:**
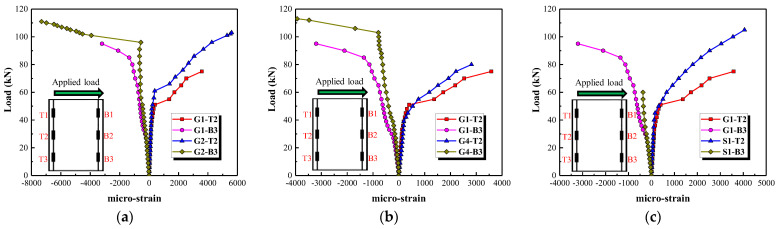
The effect of three parameters on the strain in the longitudinal bars. (**a**) Stirrup reinforcement ratio, (**b**) Longitudinal reinforcement ratio, (**c**) Type of stirrups.

**Figure 18 polymers-15-00567-f018:**
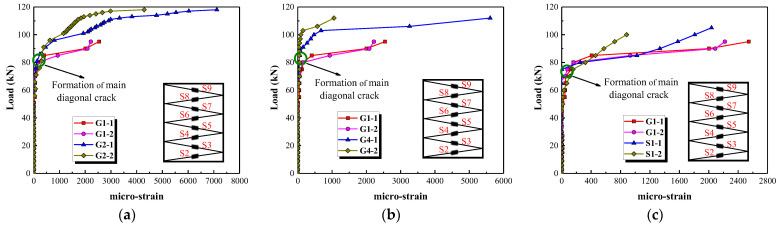
The effect of three parameters on strain in the stirrups. (**a**) Stirrup reinforcement ratio, (**b**) Longitudinal reinforcement ratio, (**c**) Type of stirrups.

**Figure 19 polymers-15-00567-f019:**
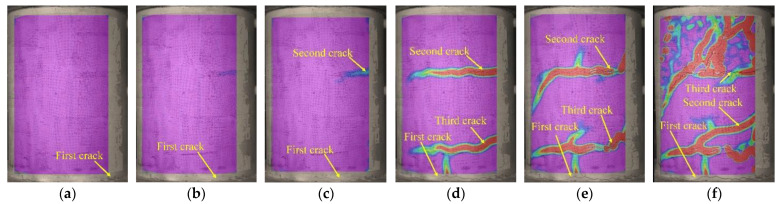
Principal strain nephograms of test specimen G1. (**a**) 24 kN, (**b**) 36 kN, (**c**) 51 kN, (**d**) 70 kN, (**e**) 80 kN, (**f**) 90 kN.

**Figure 20 polymers-15-00567-f020:**
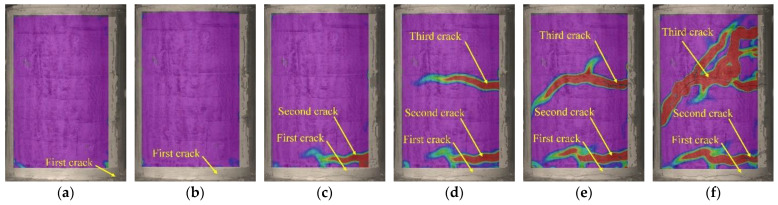
Principal strain nephograms of test specimen G3. (**a**) 33 kN, (**b**) 44 kN, (**c**) 64 kN, (**d**) 69 kN, (**e**) 80 kN, (**f**) 90 kN.

**Figure 21 polymers-15-00567-f021:**
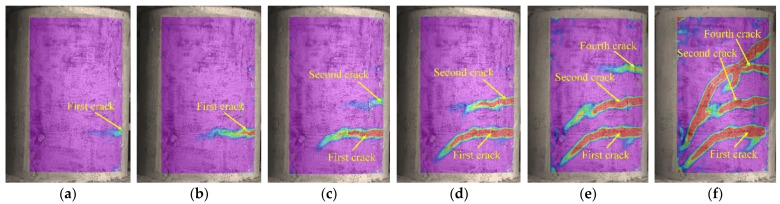
Principal strain nephograms of test specimen S1. (**a**) 27 kN, (**b**) 40 kN, (**c**) 50 kN, (**d**) 65 kN, (**e**) 80 kN, (**f**) 95 kN.

**Figure 22 polymers-15-00567-f022:**
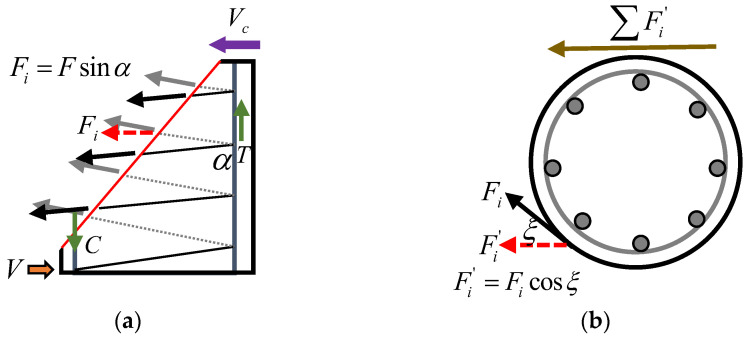
Shear transfer mechanism of a circular column with spiral stirrups. (**a**) Front view, (**b**) Top view.

**Figure 23 polymers-15-00567-f023:**
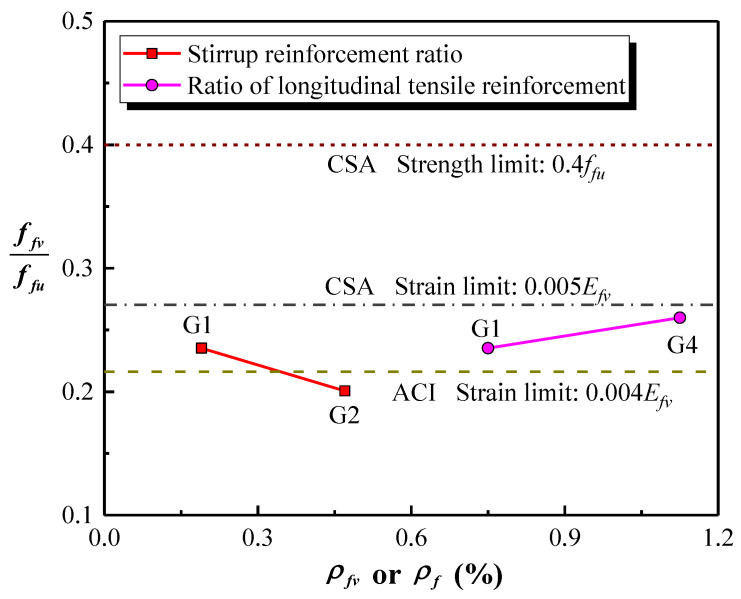
The effective stress of the spiral stirrups.

**Figure 24 polymers-15-00567-f024:**
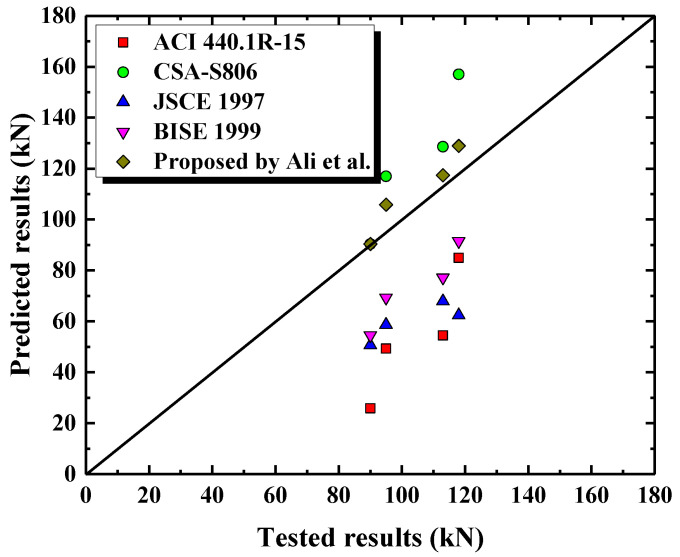
Comparison of predicted and tested shear capacities.

**Figure 25 polymers-15-00567-f025:**
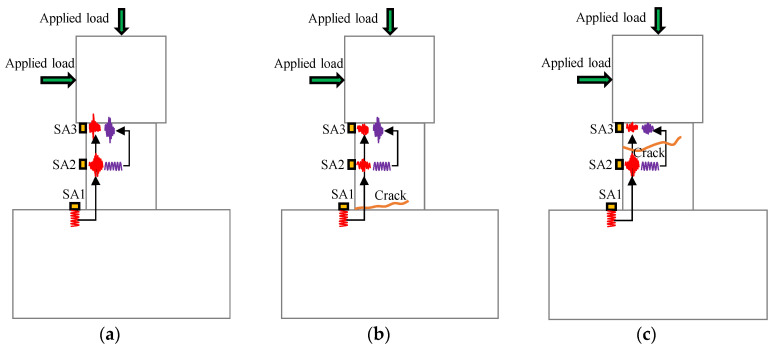
The damage detection principle of a column. (**a**) Healthy status, (**b**) Crack in the lower part, (**c**) Crack in the upper part.

**Figure 26 polymers-15-00567-f026:**
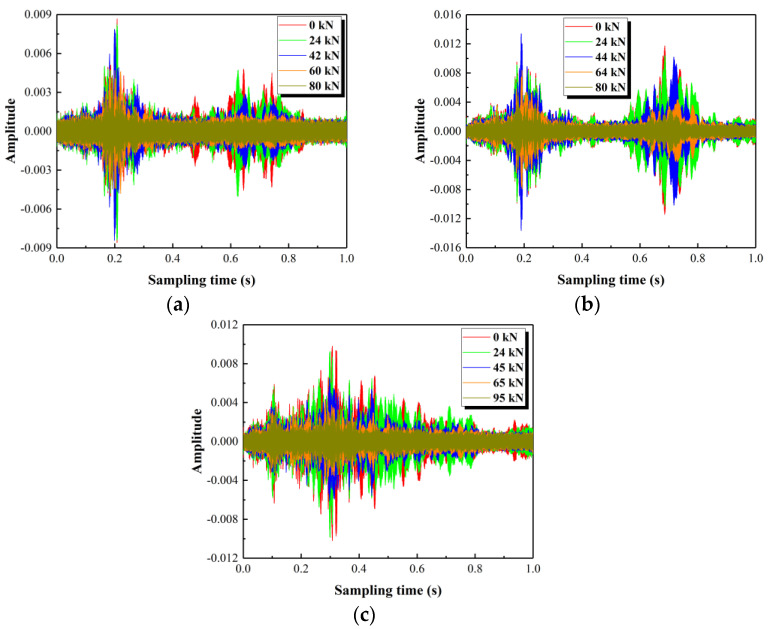
Time-domain signals of the test specimens. (**a**) Test specimen G1, (**b**) Test specimen G3, (**c**) Test specimen S1.

**Figure 27 polymers-15-00567-f027:**
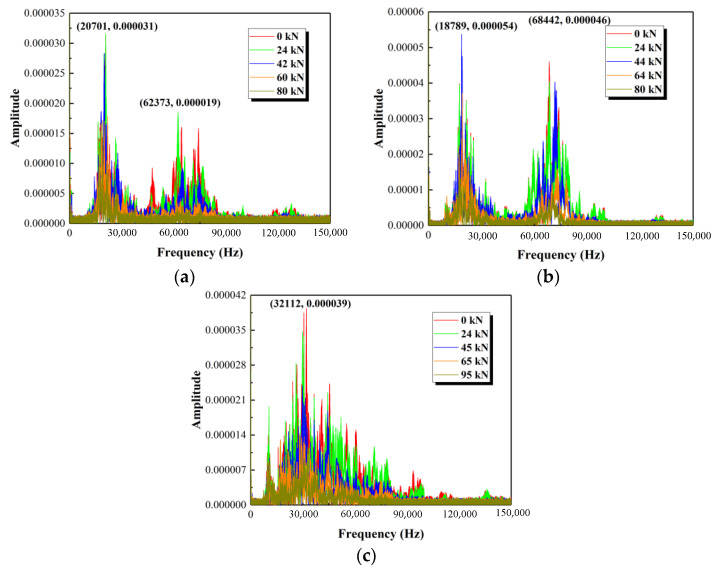
Frequency-domain diagrams for three test specimens. (**a**) Test specimen G1, (**b**) Test specimen G3, (**c**) Test specimen S1.

**Figure 28 polymers-15-00567-f028:**
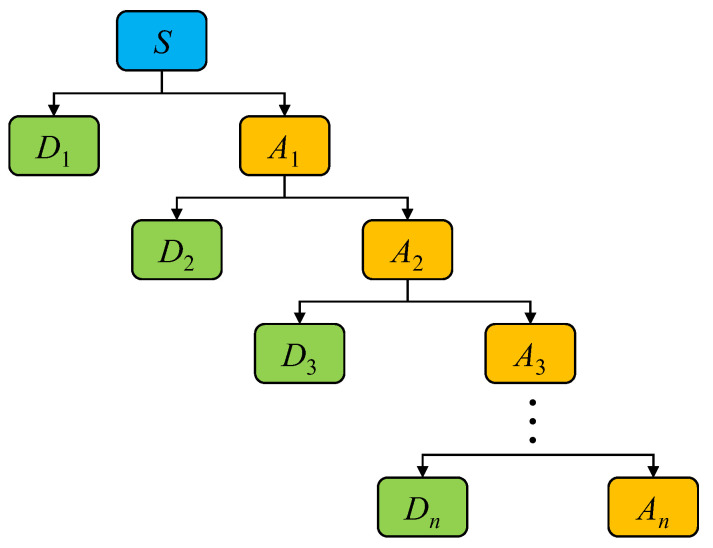
Wavelet decomposition.

**Figure 29 polymers-15-00567-f029:**
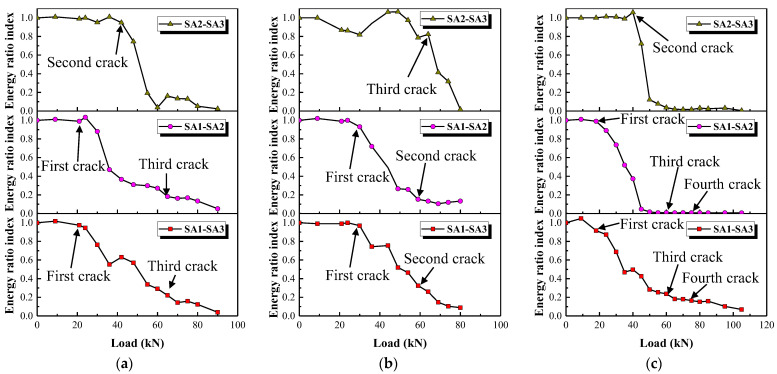
*ERI* values of three test specimens. (**a**) Test specimen G1, (**b**) Test specimen G3, (**c**) Test specimen S1.

**Figure 30 polymers-15-00567-f030:**
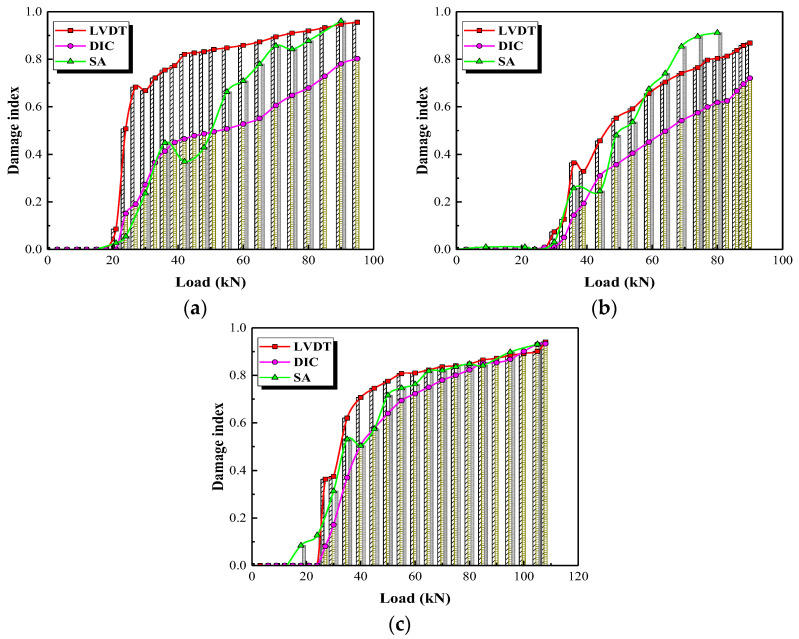
Damage levels of the test specimens. (**a**) Test specimen G1, (**b**) Test specimen G3, (**c**) Test specimen S1.

**Table 1 polymers-15-00567-t001:** Mechanical properties of reinforcing materials.

Type	Diameter (mm)	Area (mm^2^)	Surface Treatment	Yield Strength (MPa)	Tensile Strength (MPa)	Elastic Modulus (GPa)	Fracture Strain (%)
A	13	132.7	SWSC	-	1047.2	57.9	1.81
B	6	28.3	SWSC	-	860.0	46.5	1.85
C	6	28.3	-	444.3	483.2	210.0	-

Note: A, B and C represent GFRP longitudinal bar, GFRP spiral stirrup and round steel stirrup, respectively. SWSC stands for spiral wound and sand coated.

**Table 2 polymers-15-00567-t002:** Detailed parameters of concrete short columns.

Name	Shear-Span Ratio	Type of Longitudinal Bars	Type of Stirrups	Longitudinal Reinforcement Ratio (%)	Stirrup Reinforcement Ratio (%)
G1	1.33	GFRP	GFRP	1.50	0.19
G2	1.33	GFRP	GFRP	1.50	0.47
G3	1.33	GFRP	-	1.50	0.00
G4	1.33	GFRP	GFRP	2.25	0.19
S1	1.33	GFRP	Round steel	1.50	0.19

Note: G and S represent GFRP spiral stirrup and round steel stirrup, respectively.

**Table 3 polymers-15-00567-t003:** Test results of all test specimens.

Name	The Load Corresponding to Crack Occurrence (kN)				Shear Capacity (kN)	Ultimate Displacement (mm)	Ratio of Cracking Load to Shear Capacity (%)	Failure Mode
	First Crack	Second Crack	Third Crack	Fourth Crack				
G1	24	51	70	-	95	10.4	25.3	Shear-compression
G2	36	56	61	-	118	31.0	30.5	Shear-compression
G3	33	59	69	-	90	6.2	36.7	Shear-compression
G4	36	42	60	85	113	15.6	31.9	Shear-compression
S1	24	50	65	80	108	12.2	22.2	Shear-compression

**Table 4 polymers-15-00567-t004:** Design equations for calculating shear capacity.

Code	Shear Design Equation	
	$V_{c}$	$V_{f}$
ACI 440.1R-15 [21]	$V_{c}=0.4\sqrt{f_{c}^{\prime }}b_{w}(kd)$ $k=\sqrt{2n_{f}\rho_{f}+{\left(n_{f}\rho_{f}\right)}^{2}}-n_{f}\rho_{f}$ $n_{f}=E_{f}/E_{c}$	$V_{f}=\frac{A_{fv}f_{fv}d}{s}(\sin\alpha )$
CSA-S806 [22]	$V_{c}=0.05\lambda k_{a}k_{s}k_{m}k_{r}{\left(f_{c}^{\prime }\right)}^{1/3}b_{w}d_{v}$ $1\leq k_{a}=2.5Vd/M\leq 2.5$ $k_{s}=750/\left(450+d\right)\leq 1$ $k_{m}={\left(Vd/M\right)}^{1/2}\leq 1$ $k_{r}=1+{\left(E_{f}\rho_{f}\right)}^{1/3}$	$V_{f}=A_{fv}f_{fv}d_{v}\cot\theta /s$ $d_{v}=0.9d$
JSCE 1997 [23]	$V_{c}=0.2\sqrt[{4}]{1000/d}\sqrt[{3}]{100\frac{A_{f}}{b_{w}d}\frac{E_{f}}{E_{s}}}\sqrt[{3}]{f_{c}^{\prime }}b_{w}d$	$V_{f}=\frac{A_{fv}E_{fv}\varepsilon_{fvd}\left(\sin\alpha +\cos\alpha \right)}{s}z$ $\varepsilon_{fvd}=10^{-4}\sqrt{{\left(\frac{h}{300}\right)}^{-\frac{1}{10}}f_{c}^{\prime }\frac{\rho_{f}E_{f}}{\rho_{fv}E_{fv}}}$ z=d/1.15
BISE 1999 [24]	$V_{c}=0.79{\left(\frac{100\rho_{f}E_{f}}{E_{s}}\right)}^{1/3}{\left(\frac{400}{d}\right)}^{1/4}{\left(\frac{f_{c}^{\prime }}{25}\right)}^{1/3}b_{w}d$	$V_{f}=\frac{0.0025E_{fv}A_{fv}d}{s}$

Note: $V_{c}$ and $V_{f}$ are the shear strength provided by concrete and by FRP spiral stirrups, respectively; $f_{c}^{\prime }$ is the axial compressive strength; α is the angle of inclination of the spiral stirrups; $b_{w}$ refers the width of a section; $b_{w}=D$ for circular concrete members. $E_{f}$, $E_{fv}$ and $E_{s}$ are the elasticity modulus of the longitudinal reinforcement, the spiral stirrups and steel bars, respectively. $\rho_{f}$ and $\rho_{fv}$ are the longitudinal tensile reinforcement and the stirrup reinforcement ratios, respectively.

**Table 5 polymers-15-00567-t005:** Comparison of predicted and tested shear capacities.

Name	ACI 440.1R-15			CSA-S806			JSCE 1997			BISE 1999			Proposed by Ali et al.		
	$V_{c}\;(\mathrm{kN})$	$V_{f}\;(\mathrm{kN})$	$\frac{V_{exp}}{V_{pred}}$	$V_{c}\;(\mathrm{kN})$	$V_{f}\;(\mathrm{kN})$	$\frac{V_{exp}}{V_{pred}}$	$V_{c}\;(\mathrm{kN})$	$V_{f}\;(\mathrm{kN})$	$\frac{V_{exp}}{V_{pred}}$	$V_{c}\;(\mathrm{kN})$	$V_{f}\;(\mathrm{kN})$	$\frac{V_{exp}}{V_{pred}}$	$V_{c}\;(\mathrm{kN})$	$V_{f}\;(\mathrm{kN})$	$\frac{V_{exp}}{V_{pred}}$
G1	25.9	23.4	1.93	90.4	26.6	0.81	50.7	8.0	1.62	54.5	14.8	1.37	90.4	15.4	0.90
G2	25.9	59.1	1.39	90.4	66.6	0.75	50.7	11.7	1.89	54.5	37.0	1.29	90.4	38.5	0.92
G3	25.9	-	3.47	90.4	-	1.00	50.7	-	1.78	54.5	-	1.65	90.4	-	1.00
G4	31.1	23.4	2.0	102.0	26.6	0.88	58.1	9.8	1.66	62.4	14.8	1.46	102.0	15.4	0.96
Average	-	-	2.22	-	-	0.86	-	-	1.74	-	-	1.44	-	-	0.94
SD	-	-	0.77	-	-	0.09	-	-	0.11	-	-	0.13	-	-	0.04

## Data Availability

The data presented in this study are available on request from the corresponding author.

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
