# Peer review of "Study on Shear Behaviors and Damage Assessment of Circular Concrete Short Columns Reinforced with GFRP Bars and Spiral Stirrups"

_polymers, 2023, doi:10.3390/polym15030567_

Round 1

Reviewer 1 Report

The article "Study on Shear Behaviors and Damage Assessment of Circular Concrete Short Columns Reinforced with GFRP Bars and Spiral Stirrups" is interesting. However, a few comments are given below, 

1) The abstract should provide an overview of the proposed methods/methodology and materials with obtained results in the form of quantitative values. Therefore, it is mandatory to include the quantitative analysis results of the proposed work in the Abstract section. It would be best not to highlight other research in the abstract section. 

2) Line --> 51, "The short concrete columns are prone to shear failure under lateral loads induced by the wind" Provide relevant references for this statement. 

3) List of abbreviations and symbols is missing.

4) Reference numbering is confusing, follow the standardized procedure. 

For instance, line 65 --> "Mohamed et al. 718", line 69 --> "Ali et al. 1617"

5) Section "2.1.1 Concrete materials" explain the design strength also. 

6)  Picture of the "round steel stirrup" is missing. Provide it under Figure 1 

7) Section "2.2 Specimen design and fabrication" which one is used as a control specimen? Have you considered a specimen with normal steel reinforcement to compare the values with GFRP reinforcement columns? If not please explain it. 

8) Line -->162 "(a) Vertical view" or top view ? 

9)  "Comparison of predicted and tested shear capacities" if data represent in a graph, it is much easier to understand and read. 

Author Response

Thanks to reviewer for suggestions.

Reviewer 2 Report

Dear respected author

The present work includes results of experimental tests performed on short columns using GFRP bars. This is an exciting topic and may improve the knowledge in this regard. However, several points to be considered by the author may improve the level of the manuscript significantly. Such points are addressed as follows:

1.       When discussing the present work, use the (active or passive) present perfect or simple past tenses. Please, check the manuscript thoroughly.

2.       The introduction section should be improved.After mentioning the objectives of the study in "Introduction" section, please provide significance of this study in engineering sector. Besides, the importance of using composite materials should be included using followings: Strengthening of shear-critical reinforced concrete T-beams with anchored and non-anchored GFRP fabrics applications; Behavior of CFRP-strengthened RC beams with circular web openings in shear zones: Numerical study; Shear strengthening of reinforced concrete T-beams with anchored and non-anchored CFRP fabrics; Optimum amount of CFRP for strengthening shear deficient reinforced concrete beams; Experimental analysis of shear deficient reinforced concrete beams strengthened by glass fiber strip composites and mechanical stitches; Numerical investigation of the parameters influencing the behavior of dapped end prefabricated concrete purlins with and without CFRP strengthening; Experimental analysis of reinforced concrete shear deficient beams with circular web openings strengthened by CFRP composite; Experimental investigation of shear capacity and damage analysis of thinned end prefabricated concrete purlins strengthened by CFRP composite; Shear strengthening of reinforced concrete beams with minimum CFRP and GFRP strips using different wrapping technics without anchoring application.

3.       What is original, and what is significant about this work? Deep discussion are needed.

4.       There are many grammar mistakes. It should be revised.

5.       The selected elements are designed according to which regulation. Also, does it meet the requirement of being a column?

6.       Has the vertical load on the column been kept constant during the experiment? At what ratio of the column axial capacity is the axial normal load applied.

7.       Fig 1-5-7-8-9-10-16-17-18 must be detailed explain in the text.

8.       Provided discussions on the results is not enough.

9.       Please re-upload all figures at least 300 dpi.

10.   Include a table summarizing all test results and damages.

11.   The conclusion section needs to be re-written by incorporating general conclusions from the findings of this research. Please point out the novelty of this research article in this section. % are needed.

12.   What is the lesson learned? Please clarify before acceptance.

Author Response

Thanks to reviewer for suggestions.

Round 2

Reviewer 2 Report

Thanks to the authors for their responses. The manuscript can be accepted in its current form.